# Direct visualization of a native Wnt in vivo reveals that a long-range Wnt gradient forms by extracellular dispersal

Ariel M Pani[1,2]*, Bob Goldstein[1,2]

[1]Department of Biology, University of North Carolina at Chapel Hill, North Carolina, United States; [2]Lineberger Comprehensive Cancer Center, University of North Carolina at Chapel Hill, North Carolina, United States

**Abstract** Wnts are evolutionarily conserved signaling proteins with essential roles in development and disease that have often been thought to spread between cells and signal at a distance. However, recent studies have challenged this model, and whether long-distance extracellular Wnt dispersal occurs and is biologically relevant is debated. Understanding fundamental aspects of Wnt dispersal has been limited by challenges with observing endogenous ligands in vivo, which has prevented directly testing hypotheses. Here, we have generated functional, fluorescently tagged alleles for a *C. elegans* Wnt homolog and for the first time visualized a native, long-range Wnt gradient in a living animal. Live imaging of Wnt along with source and responding cell membranes provided support for free, extracellular dispersal. By limiting Wnt transfer between cells, we confirmed that extracellular spreading shapes a long-range gradient and is critical for neuroblast migration. These results provide direct evidence that Wnts spread extracellularly to regulate aspects of long-range signaling.
DOI: https://doi.org/10.7554/eLife.38325.001

## Introduction

Wnts are a family of secreted signaling proteins with critical roles in development, homeostasis, and disease (*Nusse and Clevers, 2017*), and anteroposterior gradients of Wnt activity are a common and ancient feature of animal development (*Darras et al., 2018*; *Petersen and Reddien, 2009*; *Harterink et al., 2011a*; *Scimone et al., 2016*; *Nordström et al., 2002*; *Kiecker and Niehrs, 2001*). Wnt proteins have often been thought to spread extracellularly to form concentration gradients and act directly at long ranges from their source cells (*Zecca et al., 1996*; *Neumann and Cohen, 1997*; *Strigini and Cohen, 2000*; *Kiecker and Niehrs, 2001*; *Rhinn et al., 2005*), but recent work has called this paradigm into question (*Alexandre et al., 2014*; *Farin et al., 2016*; *Stanganello et al., 2015*; *Huang and Kornberg, 2015*; *Stanganello and Scholpp, 2016*; *Mattes et al., 2018*). An increasingly prominent hypothesis is that Wnts do not freely spread over long distances in vivo and instead are primarily short-range signaling molecules (*Loh et al., 2016*) that are transferred at contacts between signaling and receiving cells that are either in close proximity or linked by dynamic plasma membrane extensions called cytonemes or signaling filopodia (*Stanganello et al., 2015*; *Huang and Kornberg, 2015*; *Stanganello and Scholpp, 2016*; *Farin et al., 2016*; *Mattes et al., 2018*). Additionally, migrating cells can deliver membrane-bound Wnts (*Pfeiffer et al., 2000*; *Serralbo and Marcelle, 2014*), and a Wnt gradient across a cell lineage can form by short-range ligand transfer followed by dilution of receptor-bound Wnt through cell divisions (*Farin et al., 2016*). In contrast, free, extracellular spreading has never been directly shown for an endogenous Wnt. These findings are consistent with a model where free, extracellular Wnt spreading is rare, and long-range Wnt gradients are most likely generated by cytoneme/filopodia- or cell lineage-based mechanisms.

**\*For correspondence:**
ariel.pani@gmail.com

**Competing interests:** The authors declare that no competing interests exist.

**eLife digest**  Cells exchange signaling proteins that help them to communicate with each other. These signals control which genes are active, and how cells grow and specialize to do different tasks. Signals can also help cells to position themselves inside a body to form new tissues and organs. For example, the so-called Wnt signaling system is important for many processes in the body, from early development to the growth and maintenance of tissues.

Signaling proteins are often thought to travel long distances between cells that produce them and the cells that respond to them. How these molecules move between cells has been challenging to study in a natural context. Signals may travel by diffusion – the random movement of molecules over time. But this had not been directly shown, and some studies suggest that thin, finger-like extensions from cells help carry the signals.

Pani and Goldstein investigated how Wnt signals travel between cells in the round worm, *Caenorhabditis elegans*. The Wnts were labelled with fluorescent tags inserted into the genome, which made them glow under certain lights. The results showed that Wnts can travel quickly between remote cells by using diffusion. Diffusion can create gradients of Wnt over long distances, with higher levels near the cells that produce Wnt and lower in others. When the Wnts were prevented from spreading freely across cells, they could not travel as far or act on their regular target cells. Both Wnt molecules and Wnt receptor proteins clustered on thin cell extensions in some cells, but the extensions were not necessary for helping the molecules spread.

This study helps us to understand one way that Wnt can traverse cells. A next step will be to examine if this aspect of Wnt signaling is similar between worms and humans. In humans, faulty Wnt signaling is implicated in many cancers. A better understanding of how this pathway normally works may help researchers develop ways to manipulate Wnt signaling in diseases.

DOI: https://doi.org/10.7554/eLife.38325.002

Furthermore, experiments demonstrating that an endogenously membrane-tethered Wnt is sufficient for many aspects of patterning and growth control in flies (*Alexandre et al., 2014*) indicated that Wnt spreading between cells is not necessarily required for proper development in some contexts where Wnt protein gradients normally occur. However, basic aspects of Wnt dispersal and the existence and functions of free, extracellular spreading have been debated without the most direct types of evidence – visualization of endogenous Wnts in vivo and methods for limiting Wnt transfer between cells without altering intrinsic spreading ability.

Two major challenges for investigating how Wnts disperse and how Wnt gradients form are the historical inability to visualize endogenous Wnts in living animals, and difficulties in observing potential cytonemes/signaling filopodia, which are often not preserved by standard tissue fixations or marked by cytoplasmic fluorescent proteins (*Sanders et al., 2013*; *Kornberg, 2017*). The ideal system to test if Wnts spread extracellularly over long distances in vivo would allow for live imaging of endogenously tagged Wnt in concert with labeled signaling and receiving cell membranes. This type of experiment would make it possible to directly visualize how far Wnt proteins spread from source cells in vivo and the extent to which Wnts located far from their sources might be associated with cryptic cell membrane structures that could mediate contact-dependent signaling. However, for technical reasons live imaging experiments on Wnt gradient formation have relied on exogenously delivered fusion proteins, with their associated caveats, and Wnts have been challenging to fluorescently tag (*Willert and Nusse, 2012*). As a consequence, there are extremely limited data on Wnt localization in vivo, and the existence and potential roles of free, extracellular Wnt spreading are unresolved. Here, using biologically functional, fluorescently-tagged knock-in alleles for an endogenous Wnt, we report the first in vivo visualization of a native long-range Wnt gradient, provide evidence for free, extracellular Wnt dispersal, and demonstrate that free, extracellular spreading is required for aspects of long-range Wnt signaling in vivo.

## Results

### In vivo visualization of a native Wnt gradient

Based on our recently-demonstrated ability to generate a functional, mNeonGreen (mNG) knock-in tag for a *C. elegans* Wnt homolog (*mom-2*) (*Heppert et al., 2018*) that is required for embryonic viability and endoderm specification at the 4 cell stage, we reasoned that *C. elegans* might be a tractable system to pursue the question of how native Wnts disperse in an animal amenable to in vivo imaging. We focused our attention on the Wnt homolog *egl-20*, which is expressed in a cluster of posterior cells (*Harterink et al., 2011a*; *Whangbo and Kenyon, 1999*), forms an anteroposterior gradient in transgenic experiments (*Whangbo and Kenyon, 1999*; *Coudreuse et al., 2006*), and regulates cell migrations, fates, polarities, and axon guidance along the anteroposterior axis during embryonic and larval development (*Harris et al., 1996*; *Maloof et al., 1999; Whangbo and Kenyon, 1999; Coudreuse et al., 2006; Prasad and Clark, 2006; Pan et al., 2006; Green et al., 2008; Yamamoto et al., 2011*; *Harterink et al., 2011a*; *Mentink et al., 2014*). Previous work has shown that Wnt/EGL-20 processing and secretion relies on many of the same factors as Wnts in flies and vertebrates (*Coudreuse et al., 2006; Bänziger et al., 2006; Harterink et al., 2011b*), making it a useful paradigm to investigate mechanisms of long-range Wnt movement that are likely to be relevant to other animals. Here, we have engineered fluorescently-tagged, biologically functional knock-in alleles for Wnt/EGL-20 using Cas9-triggered homologous recombination (*Dickinson et al., 2015*). We inserted the amphioxus-derived fluorescent protein mNG (*Shaner et al., 2013*) or the GFP/YFP derivative mYPET (hereafter YPET) at the C-terminus of Wnt/*egl-20* similar to how we tagged Wnt/*mom-2*. EGL-20::mNG and EGL-20::YPET knock-in worms had normal external morphologies and did not exhibit the characteristic defects in Q neuroblast migration seen in *egl-20* mutants (*Harris et al., 1996*; *Whangbo and Kenyon, 1999*)(*Figure 1a,b*). Subsequently, mNG and YPET tagged strains were used interchangeably depending on the relative importance of fluorescent protein photostability (mNG) versus brightness/signal:noise ratio (YPET) in different experiments (*Heppert et al., 2016*).

We first observed tagged Wnt/EGL-20 fluorescence in posterior cells in comma-stage embryos along with isolated punctae in more anterior regions (*Figure 1c,d*) and found that tagged Wnt spread from the posterior to the mid-body region of the embryo within 60 min (*Figure 1d*). In early larvae, tagged Wnt/EGL-20 formed an anteroposterior gradient along approximately the posterior half of the worm (*Figure 1e–h*) with low levels of protein detectable along the entire body axis (*Figure 1f,h*). Tagged Wnt localized most conspicuously to posterior cells near its source along with longitudinal body wall muscles, epithelial seam cells, neuroblasts, and ventral midline cells (*Figure 1e,g*; *Figure 1—figure supplements 1* and *2*). Because the mid-body seam cell precursors and most body wall muscles are already present and positioned (*Sulston et al., 1983*) by the time tagged Wnt/EGL-20 was first visible in comma-stage embryos, we can rule out the prospect that cell lineage-dependent mechanisms generate the observed anteroposterior gradient, which suggests that Wnt can move across multiple cells in this context.

### Wnt/EGL-20 localization and cell membrane architectures suggest extracellular Wnt dispersal

To determine the locations, shapes, and behaviors of Wnt/*egl-20*-expressing cells in vivo, we used the *egl-20* upstream intergenic region (hereafter P*egl-20*>) to drive a single copy plasma membrane reporter consisting of 2 copies of mKate2 fused to a PH domain from PLC1Δ1, a plasma membrane marker previously used to visualize intricate cellular architectures in *C. elegans* (*Linden et al., 2017*). In early larvae, this reporter was expressed in a cluster of posterior cells including rectal epithelial cells, the overlying dorsal and ventral body wall muscles, the stomatointestinal muscles, the anal depressor muscle, and P11/12, along with weak expression in several head neurons (*Figure 1g*, *Figure 1—figure supplement 2*), which is largely consistent with smFISH data on *egl-20* transcript localization (*Harterink et al., 2011a*) and previous transgenes (*Whangbo and Kenyon, 1999*). This reporter also labeled several posterior neurons and their projections along the ventral nerve cord that terminated in the nerve ring (*Figure 1g*; *Figure 1—figure supplement 2*). Tagged Wnt protein clearly localized near reporter-labeled axons in the head (*Figure 1—figure supplement 2*), suggesting they could act as local sources of Wnt for ventral and head cells separately from the overall

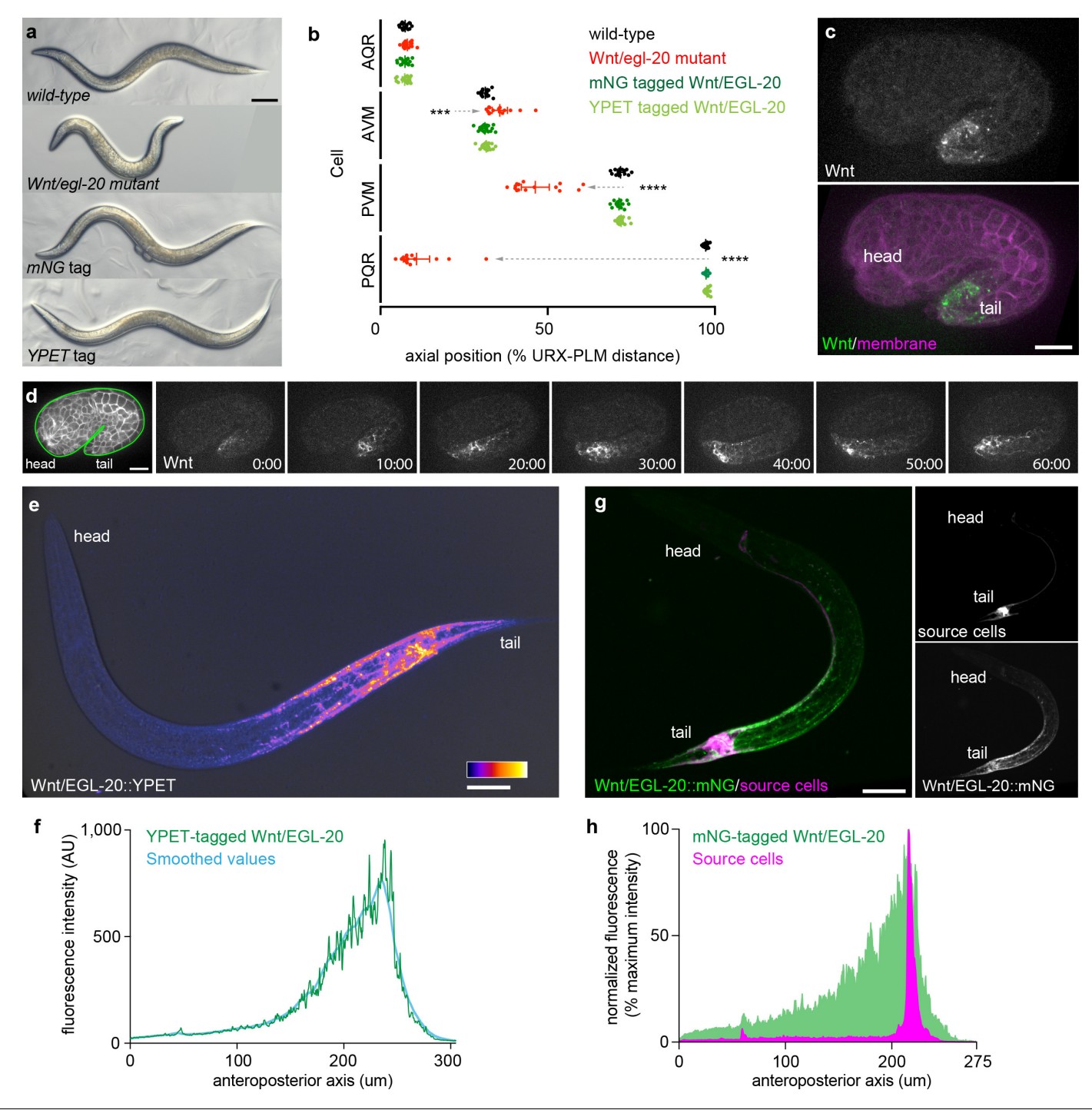

**Figure 1.** Tagged Wnt/EGL-20 is biologically functional and forms a long-range, anteroposterior gradient in vivo. (**a**) transmitted light images of adult *C. elegans* with wild-type *egl-20*, the *egl-20* loss-of-function mutant egl-20(n585), *mNG*-tagged *egl-20*, or *YPET*-tagged *egl-20* showing normal external anatomy in *mNG* and *YPET*-tagged strains; (**b**) positions of QR neuroblast descendants AQR and AVM and QL neuroblast descendants PVM and PQR in wild-type, *egl-20* mutant, *egl-20::mNG*, and *egl-20::YPET* strains showing that tagged EGL-20 is biologically functional for Q neuroblast migration. Dashed arrows indicate abnormal cell migrations. Means and 95% confidence intervals are shown for each cell type/genotype. Wild-type n = 15, egl-20 (n585) n = 15, EGL-20::mNG n = 20, EGL-20::YPET n = 18.***, adjusted p=0.0005; ****, adjusted p<0.0001, all other comparisons adjusted p>0.9999, one-way ANOVA with Sidak's multiple comparisons test; (**c**) maximum intensity projection of a comma stage embryo showing the earliest detectable Wnt/EGL-20::mNG fluorescence; (**d**) surface optical sections from time-lapse images of Wnt/EGL-20::mNG showing the onset of spreading from 1.5-fold to 2-fold stages; (**e**) maximum intensity projection of Wnt/EGL-20::YPET fluorescence in a living, late L1 stage animal illustrating the anteroposterior

*Figure 1 continued on next page*

*Figure 1 continued*

Wnt gradient colored with fire look-up-table and overlaid with transmitted light image; (f) profile plot of raw and LOWESS smoothed Wnt/EGL-20::YPET fluorescence intensity along the anteroposterior axis in the same worm as in (e); (g) maximum intensity projections of a living, mid L1 stage animal showing plasma membranes of *egl-20* source cells labeled by P*egl-20*>2x mKate2::PH (magenta) and Wnt/EGL-20::mNG protein (green); (h) profile plot of normalized EGL-20::mNG and P*egl-20*>2x mKate2::PH fluorescence intensities along the anteroposterior axis illustrating Wnt dispersal from source cells. Images are oriented with anterior to left and dorsal to top. Scale bars = 0.1 mm in a, 10 μm in c and d, and 20 μm in e and g.

DOI: https://doi.org/10.7554/eLife.38325.003

The following source data and figure supplements are available for figure 1:

**Source data 1.** Positions of Q neuroblast descendants in wild type, egl-20(n585), EGL-20::mNG, and EGL-20::YPET strains.

DOI: https://doi.org/10.7554/eLife.38325.007

**Source data 2.** Fluorescence intensity values for EGL-20::YPET, EGL-20::mNG, and Pegl-20 > 2 x mKate2::PH.

DOI: https://doi.org/10.7554/eLife.38325.008

**Figure supplement 1.** Tissue-specific Wnt/EGL-20::mNG localization.

DOI: https://doi.org/10.7554/eLife.38325.004

**Figure supplement 2.** Wnt/EGL-20 localization and plasma membrane architectures of source cells.

DOI: https://doi.org/10.7554/eLife.38325.005

**Figure supplement 3.** Visualization of the native *egl-20*-expressing cell lineage in vivo.

DOI: https://doi.org/10.7554/eLife.38325.006

anteroposterior gradient. Despite extensive attempts, we did not observe extensions from *Pegl-20*-expressing cells resembling cytononemes/signaling filopodia. Although the stomatointestinal muscles have an elaborate shape including muscle arms that extend dorsally (*Figure 1—figure supplement 2*), this morphology is similar throughout development, and we did not observe dynamic behaviors of these cells. Comparing membrane reporter and tagged Wnt fluorescence intensity along the anteroposterior axis indicated that the protein formed a gradient emanating from the posterior source cells consistent with extracellular spreading (*Figure 1h*).

To ensure that we detected all cells that expressed *egl-20* at any time of development - and therefore to assess the possibility of autocrine signaling - we marked the lineage of cells that expressed *egl-20* by inserting *flp::F2A* upstream of endogenously tagged *egl-20::mNG* to engineer a bicistronic gene expressing both a FLP recombinase and tagged EGL-20 from its native locus (*Figure 1—figure supplement 3a*). We combined this tool with a ubiquitously-expressed transgene that irreversibly converts a membrane marker from red to cyan after excision by FLP (*Figure 1—figure supplement 3a*). This experiment confirmed our interpretation of the reporter transgene pattern including the previously undescribed neuronal expression and lack of expression in anterior body wall muscles (*Figure 1—figure supplement 3b,c*) that sometimes showed weak expression of P*egl-20*-driven transgenes. These results demonstrate that broad, unseen expression at earlier stages does not underlie the post-embryonic EGL-20 distribution, and the results argue against the possibility of an autocrine/cellular memory mechanism for EGL-20 signaling as was described for Wnt signaling in the *Drosophila* wing disc (*Alexandre et al., 2014*).

Next, to test if cells known to respond to Wnt/EGL-20 possess cellular extensions that could directly mediate signaling, we used plasma membrane markers to investigate membrane architectures of Q neuroblasts, which migrate under the control of Wnt/EGL-20 signaling during early larval development (*Harris et al., 1996*; *Whangbo and Kenyon, 1999*; *Maloof et al., 1999*). In vivo imaging of these cell membranes in concert with Wnt/EGL-20 source cell membranes and tagged ligand showed that neither Wnt producing cells nor responding neuroblasts produced structures that could directly support contact-dependent signaling between them during Q neuroblast polarization through the early stages of migration (*Figure 2a–c*). We did observe that tagged Wnt/EGL-20 localized to posteriorly directed, lamellipodia-like protrusions and short filopodia on polarizing QL neuroblasts (*Figure 2a*, *Figure 3a,b*) that shortly afterwards undergo a stereotypical posterior migration regulated by EGL-20 (*Harris et al., 1996*; *Whangbo and Kenyon, 1999*; *Maloof et al., 1999*). As a natural test of whether these protrusions might be required to capture Wnt/EGL-20, we also examined tagged Wnt localization in QR neuroblast descendants. The QR and QL neuroblasts initially have identical axial positions, and the descendants of both cells respond to Wnt/EGL-20 (*Whangbo and Kenyon, 1999*; *Mentink et al., 2014*). However, QR neuroblasts migrate towards the anterior and do not make posteriorly directed protrusions (*Middelkoop and Korswagen, 2014*).

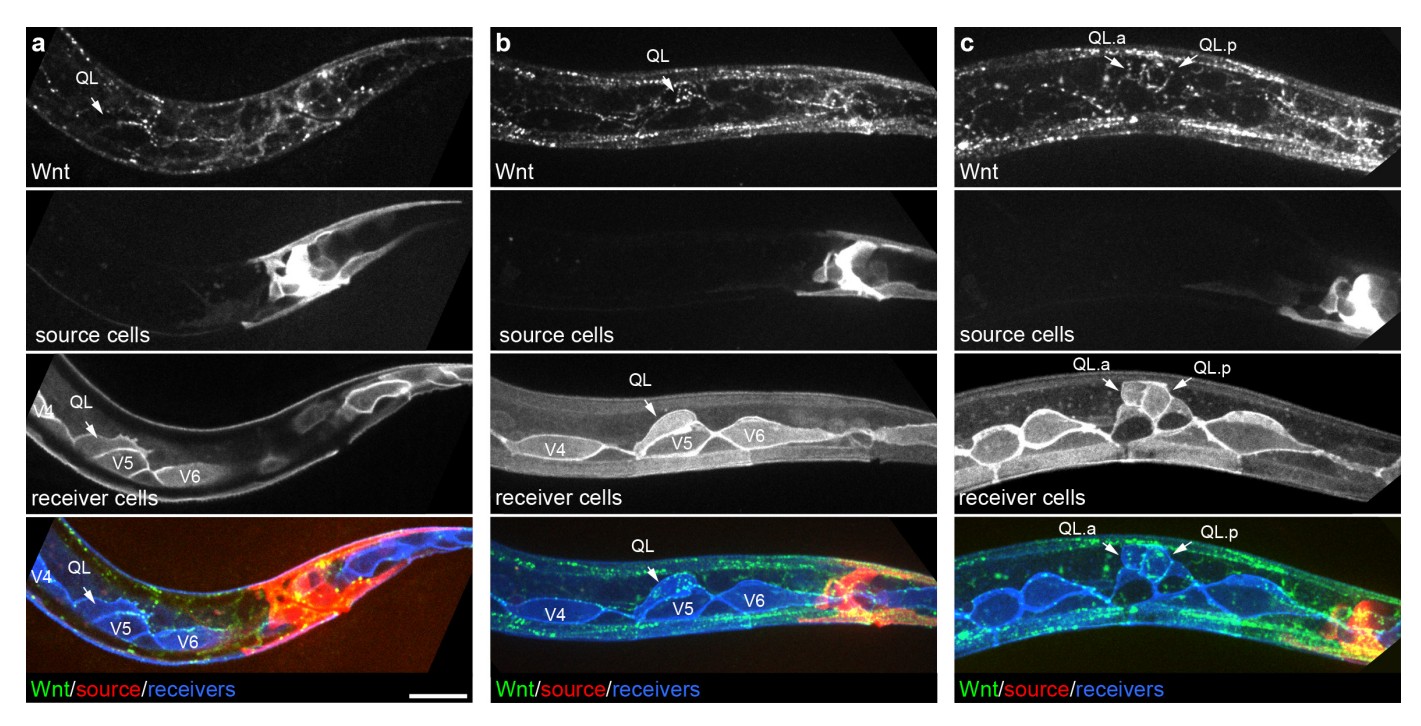

**Figure 2.** Endogenously tagged Wnt/EGL-20 localizes to responding QL neuroblasts that do not directly contact Wnt source cells. (a–c) Maximum intensity projections showing Wnt/EGL-20::mNG protein (green), plasma membranes of EGL-20 source cells marked by *Pegl-20>2x* mKate2::PH (red), and responding Q neuroblast and seam cell membranes marked by P*wrt-2>2x* mTurq2 (blue) in L1 larvae during QL polarization (a), early QL migration over the seam cell V5 (b) and the onset of QL descendant migration after the first cell division (c). Images are oriented with anterior to left and dorsal to top, scale bar = 10 μm.

DOI: https://doi.org/10.7554/eLife.38325.009

The following figure supplement is available for figure 2:

**Figure supplement 1.** Wnt/EGL-20::mNG localization in QR neuroblast descendants and seam cells.

DOI: https://doi.org/10.7554/eLife.38325.010

We found that tagged Wnt/EGL-20 also localized to QR descendants (*Figure 2—figure supplement 1a,b*) suggesting that posterior protrusions are not required for Q neuroblasts to capture and accumulate Wnt. We also found that tagged Wnt/EGL-20 localized to the posterior side of seam cells prior to their first division (*Figure 2a,b*, *Figure 2—figure supplement 1*), which is polarized in part by EGL-20 (*Yamamoto et al., 2011*), and to their posterior daughters at later stages (*Figure 1—figure supplement 1*).

Because receptor localization might provide additional insights into where Wnt signals are received in cells, we endogenously tagged two Frizzled homologs, MIG-1 and LIN-17, that are involved in QL neuroblast migration (*Harris et al., 1996*) using a design previously used to tag the Frizzled homolog MOM-5 (*Heppert et al., 2018*), and imaged them in concert with tagged Wnt/EGL-20 and labeled receiving cells in live animals. We did not observe defects in cell migrations or egg-laying in animals with tagged MIG-1 nor vulval or tail abnormalities in animals with tagged LIN-17, suggesting these fusions are biologically functional. During QL neuroblast polarization, we observed tagged Frizzled punctae that overlapped with tagged Wnt/EGL-20 and sometimes localized to short, dorsally oriented filopodia and broad protrusions from the QL neuroblast prior to its posterior migration (*Figure 3a,b*). We found that a tagged Dishevelled homolog also localized to QL neuroblast protrusions (*Figure 3—figure supplement 1*), consistent with them being a subcellular site of Wnt activation. While these findings imply that some neuroblast protrusions contribute to receiving the Wnt signal, the observed protrusions did not contact *egl-20*-expressing cells or resemble the thin and dynamic filopodia/cytonemes proposed to mediate Wnt signaling elsewhere (*Stanganello et al., 2015*; *Huang and Kornberg, 2015*).

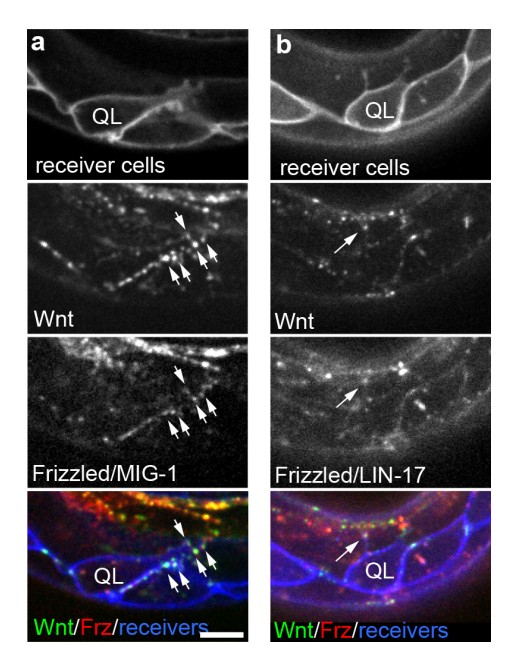

**Figure 3.** Endogenously tagged Wnt/EGL-20 and receivers co-localize on QL neuroblast protrusions. (a, b) Endogenously tagged Wnt/EGL-20::mNG (green) and Frizzled/MIG-1::mKate2 (red) (a) or Frizzled/LIN-17:: mScarlet-I (red) (b) punctae overlap and localize to protrusions from QL neuroblasts marked by P*wrt-2*>2x mTurq2::PH (blue) prior to their posterior migration. Arrows indicate punctae containing tagged Wnt and Frizzled on neuroblast protrusions; Images are oriented with anterior to left and dorsal to top, scale bar = 10 µm.

DOI: https://doi.org/10.7554/eLife.38325.011

The following figure supplements are available for figure 3:

**Figure supplement 1.** Endogenously tagged Dishevelled/MIG-5 localizes to QL neuroblast protrusions.

DOI: https://doi.org/10.7554/eLife.38325.012

**Figure supplement 2.** Extensive overlap between Wnt punctae and two Frizzled homologs in an early L1 animal.

DOI: https://doi.org/10.7554/eLife.38325.013

**Figure supplement 3.** Extensive overlap between Wnt punctae and two Frizzled homologs in a late L1 animal.

DOI: https://doi.org/10.7554/eLife.38325.014

Based on the overlap between Wnt/EGL-20 punctae and Frizzled receptors in responding Q neuroblasts, we wondered whether the Wnt punctae we observed more generally represented receptor-associated molecules, therefore most likely associated with individual receiving cells, versus some type of Wnt-containing particle that could be freely transferred between cells. Live imaging of Wnt/EGL-20::mNG simultaneously with tagged Frizzled/MIG-1 and Frizzled/LIN-17 revealed a striking overlap between tagged Wnt punctae and receptors within their respective expression domains, and most, if not all, tagged Wnt punctae visible by spinning disc confocal microscopy overlapped with one or both of the tagged receptors (*Figure 3—figure supplement 2*, *Figure 3—figure supplement 3*). These results suggest that the vast majority of bright Wnt punctae visible in our experiments most likely represent receptor-bound molecules, suggesting to us that the fraction of Wnt capable of moving between cells is present in a more diffuse form that was not easily discernible by eye.

We next sought to test if patterns of local cell-cell contacts or tissue continuity control long-range Wnt/EGL-20 spreading by investigating where Wnt secreted from specific cells was localized and it if was spatially restricted or spread broadly. To map dispersed Wnt molecules to their source cells, we tagged endogenous Wnt/EGL-20 with [FRT]mKate2::stop[FRT]mNG to generate a strain that converts irreversibly from a red to yellow fluorescent protein tag after excision by FLP recombinase (*Figure 4a*). Then, to visualize both total Wnt/EGL-20 and specific Wnt molecules derived from only a small population of cells, we used an enhancer from the posterior Hox gene *egl-5* to drive FLP expression in a small number of cells and imaged EGL-20::mKate2 and EGL-20::mNG in larval animals. These experiments showed that mNG-tagged Wnt spread from excised cells over long distances (*Figure 4b–d*). We noted little difference in the distribution of mKate2- and mNG-tagged ligands (*Figure 4b–d*), suggesting that there is a common pool of extracellular Wnt in much of the body derived from multiple cell types. Because the *egl-5* enhancer we used does not drive FLP expression in stomatointestinal muscles or neurons, we were also able to rule out the possibility that ventral cord axons or stomatointestinal muscle arms (see *Figure 1—figure supplement 2*) are the main source of Wnt/EGL-20 in more anterior regions. To test the extent to which a Wnt can spread extracellularly from ectopic source cells with different shapes and cellular neighbors, we then misexpressed Wnt/EGL-20::mNG in farther posterior tail epidermal cells using a fragment of another Wnt promoter (*lin-44*) to ensure that the ectopically expressing cells were competent to secrete functional Wnts. Similar to the endogenous protein, Wnt/EGL-20::mNG expressed from a farther posterior source was detectable in QL neuroblast descendants, seam cells, and body wall muscle

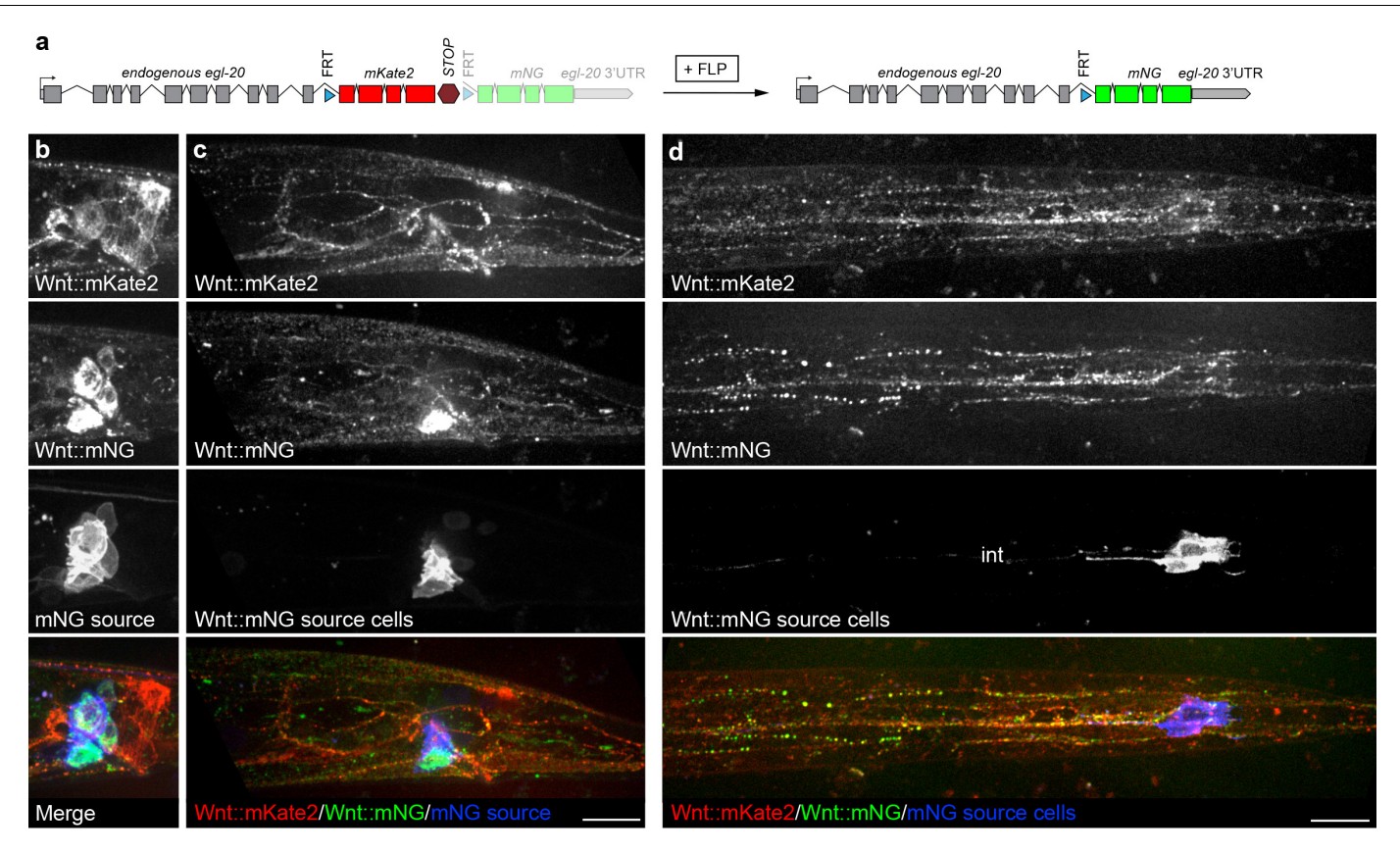

**Figure 4.** Distribution of endogenously tagged Wnt/EGL-20 secreted from different cell types suggests a common pool of extracellular ligand. (**a**) design of an endogenous *egl-20* tagged with: :$^{FRT}$mKate2::STOP$^{FRT}$mNG cassette to visibly distinguish Wnt/EGL-20 molecules produced by different cell types. By default, this allele expresses *egl-20::mKate2* followed by a stop cassette, which can be excised by FLP recombinase to irreversibly change to *egl-20::mNG*. (**b–d**) In vivo imaging of Wnt/EGL-20::mKate2 produced by non-excised cells (red), Wnt/EGL-20::mNG produced by excised cells (green), and plasma membranes of cells expressing FLP driven by an *egl-5* enhancer (blue); (**b**) maximum intensity projection of midline planes showing full shapes of cells producing Wnt/EGL-20::mKate2, Wnt/EGL-20::mNG, and FLP-expressing mNG source cells in an L2 larva; (**c**) Maximum intensity projection of surface planes in the same worm illustrating Wnt/EGL-20::mNG that has dispersed from its source cells. A region of the source cell membranes nearest the surface are visible; (**d**) maximum intensity projection of ventral surface in a separate worm showing dispersed Wnt/EGL-20::mNG from source cells. Area with isolated spots of intestinal autofluorescence in mNG source channel is labeled 'int'. Images in **b** and **c** are oriented with anterior to left and dorsal to top, scale bars = 10 µm.

DOI: https://doi.org/10.7554/eLife.38325.015

The following figure supplement is available for figure 4:

**Figure supplement 1.** Ectopically expressed Wnt/EGL-20 spreads extracellularly and localizes to native receiving cells.

DOI: https://doi.org/10.7554/eLife.38325.016

more than 75 µm and multiple cells away from source cells (*Figure 4—figure supplement 1*), suggesting that native cell contact patterns are not essential for ligand spreading to its normal target cells.

## Fluorescence recovery after photobleaching shows rapid Wnt spreading

To directly observe Wnt spreading in vivo, we used fluorescence recovery after photobleaching (FRAP), to visualize tagged Wnt/EGL-20 spreading. After photobleaching a region of interest that included a responding QL neuroblast, we found that EGL-20::YPET fluorescence began to recover in biologically relevant cells within 30 s of photobleaching (*Figure 5a–c*, *Video 1*). Performing this experiment at a stage before the QL neuroblast began its protrusive behaviors confirmed that protrusions themselves are not essential for these cells to capture Wnt (*Figure 5a,b*, *Video 1*).

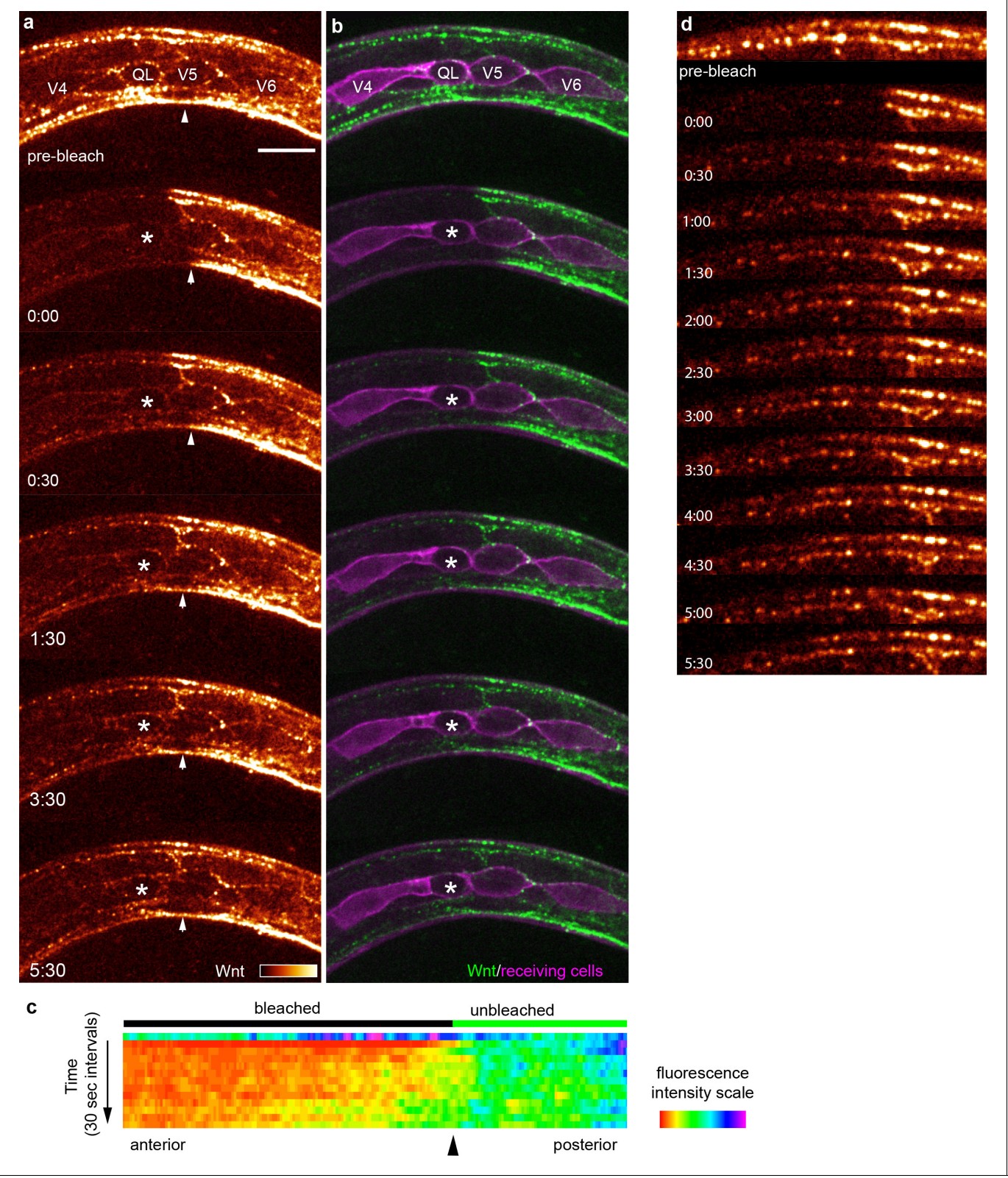

**Figure 5.** FRAP shows rapid Wnt recovery consistent with free extracellular spreading in vivo. (a, b) Images of Wnt/EGL-20::YPET fluorescence recovery at selected time points during the first 5:30 after photobleaching in a mid-body region. The bleached region of interest included a responding QL neuroblast, indicated by an asterisk, prior to its protrusive behaviors. Arrowheads indicate the anterior boundary of the unbleached region. See *Video 1* for complete time-lapse. (a) Wnt/EGL-20::YPET fluorescence colored using glow look-up-table; (b) composite images of Wnt/EGL-20::YPET
*Figure 5 continued on next page*

*Figure 5 continued*

(green) and plasma membranes of Q and seam cells (magenta). (**c**) kymograph of fluorescence intensity along the anteroposterior axis at 30 s intervals before and after photobleaching demonstrating Wnt spreading from the unbleached posterior domain; (**d**) Wnt/EGL-20::YPET fluorescence recovers in part as stable punctae suggesting dispersing Wnt molecules not individually visible by spinning disc microscopy are recruited to spatially stable clusters. Images are oriented with anterior to left and dorsal to top, scale bar = 10 μm.

DOI: https://doi.org/10.7554/eLife.38325.017

The following figure supplement is available for figure 5:

**Figure supplement 1.** Tagged Wnt recovery after photobleaching over 90 min.

DOI: https://doi.org/10.7554/eLife.38325.018

---

Interestingly, some bright Wnt punctae recovered in approximately the same locations as before photobleaching (*Figure 5d*), which suggested that mobile Wnt molecules not individually visible by spinning disc confocal microscopy in vivo were recruited to spatially stable clusters containing Wnt at the cell membrane. Recovery began rapidly but was incomplete (*Figure 5c*), even over time scales of up to 81 min (*Figure 5—figure supplement 1*, *Video 2*). This result is consistent with there being a relatively small fraction of highly mobile Wnt molecules and a larger proportion of more hindered or receptor-associated Wnts similar to what has been observed for nanoparticle-labeled FGF2 in cell culture (*Duchesne et al., 2012*) and transgenically-expressed dpp (a BMP homolog) in the *Drosophila* wing disc (*Zhou et al., 2012*). Given that tagged Wnt punctae visible by spinning disc confocal microscopy also consistently overlapped with tagged receptors in our experiments, (*Figure 3—figure supplement 2*, *Figure 3—figure supplement 3*) our data are consistent with proposed models for morphogen gradient formation based on free, extracellular movement with ligand dispersal hindered and shaped by binding to extracellular molecules including receptors (*Müller et al., 2013*).

Our findings also suggest that live imaging studies focusing on bright Wnt punctae associated with cytonemes/filopodia (*Stanganello et al., 2015*; *Huang and Kornberg, 2015*; *Holzer et al., 2012*) may not have detected a significant fraction of extracellularly mobile Wnt, which could be a general caveat for studies using live imaging of potentially diffusible signaling proteins. As a whole, our findings led us to the working hypothesis that Wnt/EGL-20 spreads between cells in the extracellular environment, and that this spreading is a key factor in gradient formation.

## Extracellular spreading between cells mediates Wnt/EGL-20 signaling

To conclusively determine whether extracellular Wnt/EGL-20 spreading shapes the endogenous anteroposterior gradient, and whether free ligand dispersal is important for normal development, we used a nanobody-based Morphotrap (*Harmansa et al., 2015*) approach to limit tagged Wnt movement between cells without otherwise altering the ligand itself. To capture and sequester extracellular Wnt/EGL-20::YPET, we used an extracellular 2x anti-GFP nanobody tethered to a CD8a transmembrane domain and intracellular mTurq2 (*Figure 6*), hereafter referred to as Morphotrap. Because ubiquitous Morphotrap expression caused intracellular accumulation of tagged Wnt in producing cells

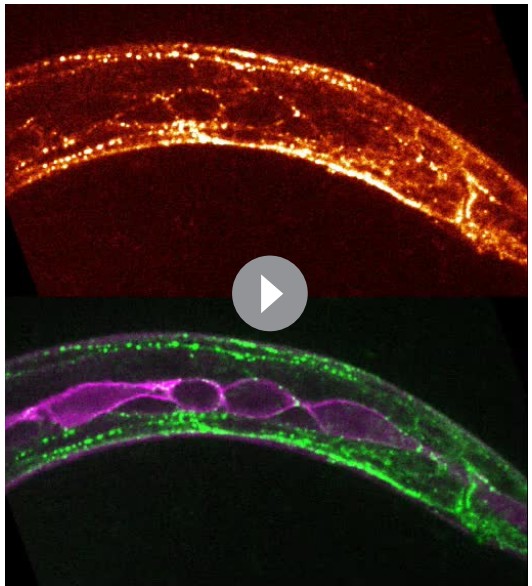

**Video 1.** FRAP experiment showing Wnt/EGL-20::YPET recovery over 5:30 in vivo. Video of time lapse images showing Wnt/EGL-20::YPET fluorescence recovery every 30 s for the first 5:30 after photobleaching in a mid-body region including seam cells and a QL neuroblast. Top panel shows Wnt/EGL-20::YPET fluorescence colored using glow look-up-table. Bottom panel shows composite of Wnt/EGL-20::YPET (green) and plasma membranes of Q and seam cells (magenta). Video corresponds with *Figure 5*.

DOI: https://doi.org/10.7554/eLife.38325.019

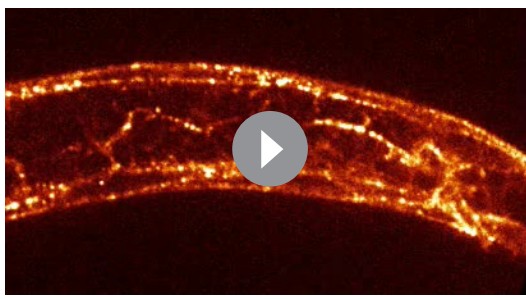

**Video 2.** FRAP experiment showing Wnt/EGL-20::YPET recovery over 81 min in vivo. Video of time lapse images showing Wnt/EGL-20::YPET fluorescence recovery every 3 min for the first 81 min after photobleaching in a mid-body region. Wnt/EGL-20:: YPET fluorescence was colored using glow look-up-table. Video corresponds with *Figure 5—figure supplement 1*.
DOI: https://doi.org/10.7554/eLife.38325.020

that suggested impaired secretion (*Figure 6— figure supplement 1*), we chose to express Morphotrap in body wall muscles using a *myo-3* promoter (*Figure 6b–d*) to allow normal Wnt release from most source cells and to prevent potentially interfering with Wnt-receptor interactions in responding neuroblasts and seam cells. Because we did not alter the intrinsic ability of Wnt/EGL-20 to spread or manipulate Wnt capture or signal transduction in known responding cells, this approach allowed us to directly test the extent to which extracellular Wnt spreading across multiple cells is required for signaling. We first found that Morphotrap expression in body wall muscles altered tagged Wnt distribution in the body, resulting in a clearly altered antero-posterior gradient shape (*Figure 6a–f*). We observed elevated levels of Wnt near source cells (*Figure 6c–e*) along with a relatively sharper decrease in levels along the anteroposterior body axis (*Figure 6f*) compared to YPET-tagged

Wnt alone. In a region anterior to Wnt/EGL-20 producing cells, Morphotrap also significantly reduced tagged Wnt levels outside of body wall muscle (*Figure 6g*, p<0.0001, Mann-Whitney test). To assess the biological relevance of this manipulation, we imaged Q neuroblast migration and observed that Morphotrap reversed QL descendant migration (*Figure 6h,i*) in a significant number of worms (n = 20/27 versus 0/23, p<0.0001, Fisher's exact test). As Morphotrap phenocopied the QL migration defects characteristic of *egl-20* loss of function mutants, we concluded that free, extracellular Wnt/EGL-20 spreading is required for signaling to QL descendants. We noticed that adult P*myo-3*>Morphotrap animals did not display the egg-laying defects seen in *egl-20* mutants, suggesting that either signaling in body wall muscles is adequate for normal development in this context or that sufficient un-trapped ligand is able to reach the essential cells, possibly as a result of Morphotrap saturation. Efforts to express Morphotrap at higher levels using multi-copy extrachromosomal arrays were unsuccessful as this frequently led to aberrant shapes and behaviors of Morphotrap-expressing cells (*Figure 6—figure supplement 1*). Together, our Morphotrap results demonstrate that: (1) a long-range Wnt gradient is shaped by extracellular spreading; (2) Wnt molecules that localize to body wall muscle and other tissues are derived from a common extracellular pool, and; (3) free, extracellular Wnt spreading between cells is required for normal neuroblast migration.

## Discussion

Taken together, our in vivo findings on endogenous Wnt localization and dispersal, along with visualization of membrane architectures of signaling and responding cells, provide the first direct evidence that a native, long-range Wnt gradient forms through free, extracellular dispersal. Our findings also challenge the idea that Wnts primarily act as short-range signals and establish that an endogenous Wnt can freely spread across multiple cells in vivo, which highlights the variety of mechanisms that animals can deploy to generate Wnt gradients. Furthermore, our Morphotrap results demonstrate that extracellular Wnt movement between cells is required for a developmental signaling event. While this result at first seems to conflict with the finding that a membrane-tethered Wnt is sufficient for essential aspects of signaling in *Drosophila*, there are key differences in Wnt expression that may explain at least some of the differential requirements for Wnt spreading in these cases. In particular, *Drosophila wingless* is broadly expressed at early third instar stages in prospective wing blade cells that are later patterned by Wnt signaling, before becoming restricted to a narrow stripe of expression at the dorsoventral boundary. The sufficiency of membrane-tethered Wingless for normal wing patterning is thought to be a consequence of this earlier autocrine signaling (*Alexandre et al., 2014*), which is not necessarily a broadly applicable mechanism. In contrast, our lineage tracing experiments using *flp::F2A* inserted at the *egl-20* locus showed that Wnt/*egl-20* is never expressed in the majority of receiving cells or their neighbors, which makes the same mechanism unlikely in *C.*

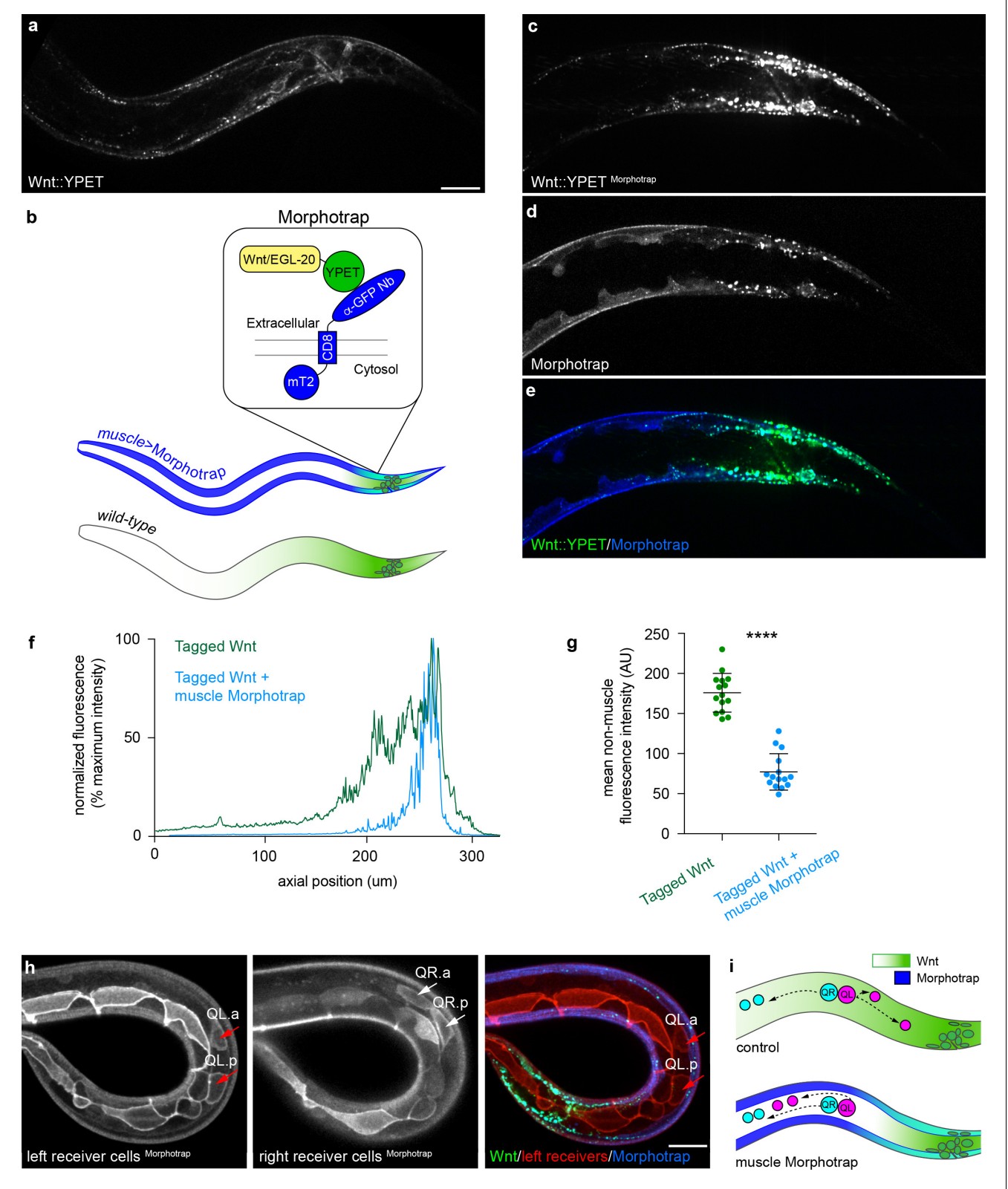

**Figure 6.** Extracellular spreading shapes long-range Wnt dispersal. (a) Normal Wnt/EGL-20::YPET fluorescence in the posterior of a late L1 larva; (b) Schematic diagram of the Morphotrap system and Wnt/EGL-20::YPET distribution in P*myo-3* >Morphotrap and control animals; (c, d) Identically acquired and processed images of Wnt/EGL-20::YPET (c, e) and P*myo-3* >Morphotrap (d, e) fluorescence in a transgenic animal showing that Morphotrap fluorescence (blue) overlaps with Wnt/EGL-20::YPET (green) and alters its spatial distribution; (f) Profile plot of normalized Wnt/EGL-20::

*Figure 6 continued on next page*

*Figure 6 continued*

YPET fluorescence intensity along the anteroposterior axis in representative animals with (blue lines) or without (green lines) Morphotrap expression in body wall muscles; (**g**) Capturing Wnt/EGL-20::YPET in body wall muscles reduces levels in adjacent tissues, p<0.0001 Mann-Whitney test. Graph shows raw data points with means and 95% confidence intervals. Control n = 15, Morphotrap n = 15; (**h**) Separate images of left and right side Q neuroblasts and seam cells in a Morphotrap transgenic animal showing reversed QL neuroblast migration. Responding cells are marked by *Pwrt*-2>2 x mKate2::PH (red), and red arrows indicate anteriorly migrating QL descendants; (**i**) schematic diagrams of Q neuroblast migration in control and Morphotrap animals showing Wnt/EGL-20::YPET distribution and migrations of QR (cyan) and QL (magenta) neuroblast descendants. QL descendants have reversed migration in Morphotrap animals and in *egl-20* mutants (see *Figure 1b*). Images in (**a–e**) are oriented with anterior to left and dorsal to top, images in (**h**) show a curled worm with tail to lower left and anterior to upper left, scale bars = 10 μm.

DOI: https://doi.org/10.7554/eLife.38325.021

The following source data and figure supplement are available for figure 6:

**Source data 1.** Fluorescence intensity values for EGL-20::YPET in control and Morphotrap animals.

DOI: https://doi.org/10.7554/eLife.38325.023

**Source data 2.** Non-muscle fluorescence intensity values for EGL-20::YPET in control and Morphotrap animals.

DOI: https://doi.org/10.7554/eLife.38325.024

**Figure supplement 1.** Intracellular Wnt-Morphotrap aggregations caused by ubiquitous Morphotrap expression and abnormal morphologies in cells with multi-copy extrachromosomal arrays.

DOI: https://doi.org/10.7554/eLife.38325.022

*elegans*. Given the evolutionarily conserved nature of the Wnt signaling pathway, our results support the idea that free, extracellular dispersal of Wnts could be a viable mechanism for long-range Wnt movement and Wnt gradient formation in many organismal contexts.

While our findings argue against a cytoneme/filopodia-based mechanism for anteroposterior Wnt gradient formation here or ligand transfer by direct contact from source cells to responding Q neuroblasts and seam cells, they do not rule out roles for cell shapes or direct ligand transfer in different situations, and we predict that the architectures of many cells have important roles in cell-cell signaling. Indeed, our finding that a tagged Wnt and its receptors colocalized on neuroblast protrusions, which also contained tagged Dishevelled punctae, is consistent with a role for these structures in receiving the Wnt signal, possibly through interactions with the extracellular matrix or nearby cells. While we have demonstrated that Wnt/EGL-20 forms a long-range gradient in vivo, our findings do not necessitate that the gradient shape is itself critical for any biological outcome. However, our results taken together with earlier experiments showing that EGL-20 overexpression can also alter neuroblast migration (*Whangbo and Kenyon, 1999*; *Mentink et al., 2014*) establish that both extracellular spreading and proper Wnt levels are required for at least some aspects of normal development.

Diffusion-based models for morphogen gradient formation in general have several perceived weaknesses (*Stanganello and Scholpp, 2016*; *Kornberg, 2017*; *Müller et al., 2013*; *Wolpert, 2016*). First, diffusion alone is predicted to be inefficient over long distances in three dimensions, and there are questions about whether it could generate a robust gradient over the time scales and distances required for developmental patterning. Second, cell-cell signaling must be tightly controlled in space and time while contending with growth and morphogenesis, and it is not clear that a passive dispersal mechanism could produce the necessary precision or stability in vivo. In light of these issues, one possibility for how animals could produce robust and spatiotemporally controlled signaling protein gradients would be for source cells to produce non-limiting amounts of ligand and for the observed tissue- and cellular-level distributions to be shaped by factors acting in receiving cells and/or the intervening extracellular environment. In this case, factors such as secreted antagonists, binding to receptors, extracellular matrix interactions, and destruction, in combination with feedback control of gene expression, might transform a relatively imprecise dispersal process such as diffusion into a robust protein gradient (*Müller et al., 2013*). One prediction of this hypothesis is that ligand abundance and spreading rates should not be the limiting factors in gradient formation, which will be important to test in the future. Signaling molecule dispersal also may not need to be actively directed to specific cells in order for accurate signaling to individual cells or tissues, and the cell-type specific and sometimes polarized Wnt/EGL-20 localization we observed in vivo suggests that responding cells themselves determine key features of signaling protein distribution and signaling activity within an organismal-scale gradient. Consistent with this possibility, cell autonomous factors

such as timed expression of different Wnt receptors in migrating cells (*Mentink et al., 2014*), autonomously-acting Syndecan-1 (*Saied-Santiago et al., 2017*), and an Rnf43/Znrf3 homolog (*Moffat et al., 2014*) are known to modulate Wnt signaling activity in cells along the *C. elegans* anteroposterior axis. The Ror/CAM-1 receptor is also thought to have dual roles in both regulating Wnt activity cell-autonomously (*Song et al., 2010*) and non-autonomously restricting ligand availability to other cells (*Green et al., 2007*), which could be a general function for other receptors as well.

Due to their hydrophobic nature (*Willert et al., 2003*), fully processed Wnts are not thought to be capable of freely moving over long distances in the extracellular space on their own (*Langton et al., 2016*), and it remains to be determined where in the extracellular environment Wnt/EGL-20 spreads and if spreading involves association with cell surface or extracellular matrix proteins (*Han et al., 2005*; *Fuerer et al., 2010*; *Mii et al., 2017*), structures such as lipoprotein particles (*Neumann et al., 2009*; *Panáková et al., 2005*) or exosomes/microvesicles (*Korkut et al., 2009*; *Gross et al., 2012*; *Greco et al., 2001*), and/or other interacting molecules to modulate Wnt solubility (*Mii and Taira, 2009*; *Mulligan et al., 2012*; *Chang and Sun, 2014*). While we have shown that extracellular dispersal is required for shaping a long-range Wnt gradient, the endogenous factors that promote and/or hinder ligand spreading await further characterization. It is tempting to speculate that mixing of extracellular fluids could also contribute to ligand spreading in vivo, although there is not currently evidence for this. Activities of cells such as protrusions, migrations, divisions, morphogenetic movements, and muscular contractions could drive extracellular mixing, potentially helping to circumvent the theoretical limitations that diffusion alone places on speed and distance of ligand dispersal. Our direct visualization of a Wnt gradient here establishes that long-range Wnt spreading can occur through free, extracellular dispersal and lays a groundwork to investigate how endogenous Wnts are transferred in vivo between cells, and how other molecules, cellular processes, and biophysical factors might orchestrate Wnt movement to control the range and strength of signaling.

## Materials and methods

### Key resources table

| Reagent type (species) or resource | Designation | Source or reference | Identifiers | Additional information |
|---|---|---|---|---|
| Strain, strain background (*C. elegans*) | cpIs89[Pwrt-2>2x mTurq2::PH::tbb-2 3'UTR loxN] I; cpIs85[Pegl-20 > 2 x mKate2::PH::3xHA::tbb-2 3'UTR loxN] II; egl-20(cp221[egl-20::mNG3xFlag]) IV | This paper | LP515 | |
| Strain, strain background (*C. elegans*) | egl-20(cp221[egl-20::mNG3xFlag]) IV; qyIs541[Pmyo-3>mCherry::PH::tbb-2 3'UTR] | This paper | LP673 | |
| Strain, strain background (*C. elegans*) | cpIs128[Pwrt2>2x mTurquoise2::PH::3xHA::tbb-2 3'UTR loxN] I; mig-1(cp360[mig-1::mKate2:3xMyc]) I; egl-20(cp221[egl-20::mNG3xFlag]) IV | This paper | LP727 | |
| Strain, strain background (*C. elegans*) | cpIs89[Pwrt-2>2x mTurquoise2::PH::tbb-2 3'UTR loxN] I; mig-5(cp385[mNG-GLO·AID::mig-5]) II | *Heppert et al., 2018* | LP728 | |
| Strain, strain background (*C. elegans*) | cpIs92 [Pmec-7>2x mTurq2::PH::3xHA::tbb-2 3'UTR loxN] I; cpIs129 [Pgcy-32 > 2 x mKate2::PH::3xHA::tbb-2 3'UTR loxN] II; egl-20(cp353[egl-20::mNG3xFlag]) IV | This paper | LP729 | |
| Strain, strain background (*C. elegans*) | cpIs92 [Pmec-7>2x mTurq2::PH::3xHA::tbb-2 3'UTR loxN] I; cpIs129 [Pgcy-32 > 2 x mKate2::PH::3xHA::tbb-2 3'UTR loxN] II | This paper | LP730 | |

*Continued on next page*

Continued

| Reagent type (species) or resource | Designation | Source or reference | Identifiers | Additional information |
|---|---|---|---|---|
| Strain, strain background (C. elegans) | cpIs130[Pwrt-2>2x mKate2 ::PH::3xHA::let-858 3'UTR::Ptag-168 > HisCl1::tbb-2 3'UTR loxN] II; egl-20 (cp353[egl-20::mNG3xFlag]) IV | This paper | LP732 | |
| Strain, strain background (C. elegans) | cpIs130[Pwrt-2>2x mKate2::PH:: 3xHA::let-858 3'UTR::Ptag-168 > HisCl1::tbb-2 3'UTR loxN] II; egl-20(cp400[egl-20::YPET3xFlag]) IV | This paper | LP783 | |
| Strain, strain background (C. elegans) | cpIs156[Pwrt2>2x mTurquoise2::PH: :3xHA::tbb-2 3'UTR SEC + loxN] I; lin-1 7(cp391[lin-17::mScarlet-C1^AID]) I; egl-20(cp221[egl-20::mNG3xFlag]) IV | This paper | LP790 | |
| Strain, strain background (C. elegans) | mig-1(cp360[mig-1::mKate23xMyc]) I; lin-17(cp404[lin-17::mTurquoise2AID]) I; egl-20(cp221[egl-20::mNG3xFlag]) IV | This paper | LP792 | |
| Strain, strain background (C. elegans) | cpIs92 [Pmec-7>2x mTurq2:: PH::3xHA::tbb-2 3'UTR loxN] I; cpIs129 [Pgcy-32 > 2 x mKate 2::PH::3xHA::tbb-2 3'UTR loxN] II; egl-20(n585) IV | This paper, CGC | LP793 | egl-20(n585) crossed to LP730 |
| Strain, strain background (C. elegans) | cpIs92 [Pmec-7>2x mTurq2:: PH::3xHA::tbb-2 3'UTR loxN] I; cpIs129 [Pgcy-32 > 2 x mKate2 ::PH::3xHA::tbb-2 3'UTR loxN] II; egl-20(cp400[egl-20:: YPET3xFlag]) IV | This paper | LP795 | |
| Strain, strain background (C. elegans) | cpIs117[Peft-3::FRT > 2 x mKate2::PH::let-858 3'UTR::FRT > 2 x mTurquoise2::PH::3x HA::tbb-2 3'UTR + loxN] I; egl-20(cp411[flp::F2A::egl-20]) IV; egl-20(cp221[egl-20::m NG3xFlag]) IV; | This paper | LP805 | |
| Strain, strain background (C. elegans) | cpIs158[Pmyo-3>pat-3sp::2x vhhGFP4:: CD8 tm::2x mTurquoise2::PH::tbb-2 3'UTR loxN] I; cpIs130[Pwrt-2>2x mKate2::PH::3xHA::let-858 3'UTR ::Ptag-168 > HisCl1::tbb-2 3'UTR loxN] II; egl-20(cp400 [egl-20::YPET3xFlag]) IV | This paper | LP815 | |
| Strain, strain background (C. elegans) | cpIs159[egl-5(K enhancer)::pes-10 delta > flp::SL2::2x mTurq2::PH::3xHA::tbb-2 3'UTR loxN] I; egl-20(cp413[egl-20::FRT5T2:: mKate2::let-858 3'UTR::FRT5T2:: mNG3xFlag]) IV | This paper | LP817 | |
| Strain, strain background (C. elegans) | cpIs160 [Plin-44 > egl-20::mNG ::SL2::2x mTurq2::PH::3x HA::tbb-2 3'UTR loxN] I; cpIs130[Pwrt-2>2x mKate2::PH::3xHA::let-858 3'UTR::Ptag-168 > HisCl1:: tbb-2 3'UTR loxN] II | This paper | LP818 | |
| Recombinant DNA reagent | Peft-3>Cas9+PU6>empty sgRNA | Dickinson et al., 2013 | pDD162 | vector for Cas9 + sgRNA cloning |
| Recombinant DNA reagent | Peft-3>Cas9+ttTi5605 sgRNA | Dickinson et al., 2013 | pDD122 | Cas9 + sgRNA targeting genomic site near ttTi5605 |
| Recombinant DNA reagent | Peft-3>Cas9+ttTi4348 sgRNA | This paper | pAP082 | Cas9 + sgRNA targeting genomic site near ttTi4348. Derived from pDD122. |

Continued

| Reagent type (species) or resource | Designation | Source or reference | Identifiers | Additional information |
|---|---|---|---|---|
| Recombinant DNA reagent | empty promoter > 2 x mKate2::PH::3xHA::tbb-2 3'UTR loxN SEC loxN ttTi5605 | This paper | pAP087.2 | vector for plasma membrane reporter insertions near ttTi5605. Derived from pCFJ150. |
| Recombinant DNA reagent | empty promoter > 2 x mTurq2::PH::3xHA::tbb-2 3'UTR loxN SEC loxN ttTi4348 | This paper | pAP088 | vector for plasma membrane reporter insertions near ttTi4348. Derived from pCFJ352. |
| Recombinant DNA reagent | mNGSEC3xFlag | Dickinson et al., 2015 | pDD268 | vector for cloning homologous repair templates |
| Recombinant DNA reagent | YPETSEC3xFlag | Dickinson et al., 2015 | pDD283 | vector for cloning homologous repair templates |
| Recombinant DNA reagent | mKate2SEC3xMyc | Dickinson et al., 2015 | pDD287 | vector for cloning homologous repair templates |

## *C. elegans* strains

*Caenorhabditis elegans* animals were cultured on Normal Growth Media (NGM) plates, fed *E. coli* (OP50 strain), and grown at 20°C for experiments. Worms were grown at 25°C for incubation during strain construction or to accelerate development during crosses. Single copy *Pwrt-2>2x mTurq2::PH* (cpIs89) or *Pwrt-2>2x mKate2::PH* (cpIs130) transgenes were used to visualize seam cells and Q neuroblasts before and during migration. Single copy *Pmec-7>2x mTurq2::PH* (cpIs92) and *Pgcy-32>2x mKate2::PH* (cpIs129) transgenes were used to visualize positions of Q neuroblast progeny after their migrations.

## Gene tagging and transgenesis

Knock-in strains were generated using Cas9-triggered homologous recombination with standard methods (*Dickinson et al., 2015*). Fluorescent protein knock-ins were made at the C-terminus for *egl-20*, *mig-1*, and *lin-17*, and *flp::F2A* was inserted directly upstream of the *egl-20* start codon. Repair templates were constructed by inserting homology arm PCR products amplified from genomic DNA into plasmids containing a fluorescent protein and self-excising selection cassette using Gibson assembly (New England Biolabs) or SapTrap assembly as described in detail elsewhere (*Dickinson et al., 2015*; *Schwartz and Jorgensen, 2016*). All C-terminal homologous repair templates included a nine amino acid flexible linker (GASGASGAS) between the endogenous coding sequence and fluorescent protein. To construct the $^{FRT}$mKate2::stop$^{FRT}$mNG cassette we used the *let-858* 3'UTR as a strong transcriptional terminator and placed FRT5T2 sites within synthetic introns between *egl-20* and *mKate2* and between the *let-858* terminator and *mNG*. Cas9 guide RNA targeting sequences for each gene were selected using the CRISPR Design Tool (*Hsu et al., 2013*) (http://crispr.mit.edu). Guide RNA target sites used were (5'−3'): *egl-20* N-terminus, GAGAATATTGCCCA TAAACG AGG; *egl-20* C terminus, GCAGTACATACATGCAAATA AGG; *lin-17* C-terminus, TCTCGC TCAGACGACCTTAC TGG; *mig-1* C-terminus, AGTTCGAAACGTCGACGCGT AGG; chromosome I near ttTi4348 site, GAAATCGCCGACTTGCGAGG AGG; chromosome II near ttTi5605 site, GATA TCAGTCTGTTTCGTAA CGG. Guide RNA sequences were cloned into the P*eft-3* >Cas9+sgRNA expression vector pDD162 using Q5 site-directed mutagenesis (New England Biolabs) and co-injected into adult germlines with repair templates and extrachromosomal array markers as described previously (*Dickinson et al., 2015*; *Dickinson et al., 2013*). Candidate knock-ins were selected by hygromycin B treatment, phenotypic identification (roller), and the absence of fluorescent extrachromosomal arrays. Candidates were singled to new plates to establish homozygous lines, and PCR genotyping was used to confirm fluorescent protein knock-ins. To excise the selectable marker cassettes, early L1 larvae were heat-shocked at 32°C for four hours to induce Cre expression, and non-roller offspring were picked in the next generation. Backbones for single copy transgene insertions using Cas9-triggered homologous recombination were made by replacing the unc-119(+) selectable marker in the MOSCI targeting vectors pCFJ150 (ttTi5605 locus) and pCFJ352 (ttTi4348 locus) with a self-excising selection cassette (*Dickinson et al., 2015*) flanked by loxN sites and deleting the guide RNA target sequences from the homology arms. Coding sequences for transgenes were codon optimized using the *C. elegans* codon adapter tool (*Redemann et al., 2011*)

(https://worm.mpi-cbg.de/codons/cgi-bin/optimize.py) and synthesized as gBlock DNA fragments (Integrated DNA Technologies). Transgene promoters were amplified from genomic DNA or existing plasmids and inserted into linearized plasmid backbones using Gibson assembly. The *egl-5* K enhancer corresponding to syEx616 (*Teng et al., 2004*) was cloned upstream of a *pes-10Δ* minimal promoter. Primers are provided in *Supplementary file 1*.

## In vivo microscopy

For imaging post-embryonic stages, larval animals were anesthetized with 0.1 mmol/L levamisole in M9 buffer (*Chai et al., 2012*) or immobilized using 0.1 μm polystyrene nanoparticles (*Kim et al., 2013*) (Polysciences) and mounted on 5–10% (wt/vol) agarose pads and maintained at room temperature (~20°C). Larval animals were imaged within one hour of mounting, and images shown in figures were representative of at least ten animals imaged on at least two occasions. Numerous animals were mounted on each slide, and animals were selected for imaging based on developmental stage, orientation on the slide, and overall health. For embryo imaging, embryos were dissected from gravid adults in egg buffer and mounted on poly-L-lysine coated coverslips with 2.5% agarose pads. Our imaging systems consisted of: (1) Nikon TiE stand with CSU-X1 spinning disk head (Yokogawa), 447 nm, 514 nm, and 561 nm solid state lasers, ImagEM EMCCD camera (Hammamatsu), iLas$^2$ (Roper Scientific), and 100 × 1.49 NA objective, or; (2) a Nikon TiE stand with CSU10 spinning disk head (Yokogawa), 514 nm and 561 nm solid state lasers, ORCA Flash sCMOS camera (Hammamatsu), and 40 × 1.3 NA or 60 × 1.4 NA objectives. Images were acquired using MetaMorph software (Molecular Devices) with a 0.3 μm step size for z-stacks and varying exposure times and laser intensities depending on the strain and developmental stage. FRAP experiments were performed using an iLas$^2$ system and software (Roper Scientific). Images were acquired immediately before and after photobleaching followed by time-lapse imaging at defined intervals. To prepare figures, image stitching was performed using FIJI where necessary. Stitched images were not used for quantitative comparisons. Images were cropped and rotated, and brightness and contrast were adjusted using FIJI (*Schindelin et al., 2012*).

## Quantification and statistical analyses

No statistical tests were used to predetermine sample size. Animals were selected for measurements based on developmental stage, orientation on the slides, and health. No animals were excluded from analyses post-hoc. Measurements from at least two imaging sessions of each worm strain were used for analyses. Results are presented as raw data points with mean and 95% confidence intervals for all graphs. P-values were considered significant at $p < 0.05$. Positions of Q neuroblast progeny after migration were quantified by using the non-motile URX neuron in the head and PLM neurons in the tail as fiducial markers. Relative positions of Q neuroblast progeny AQR, AVM, PVM, and PQR were calculated as a percentage of the distance between URX and PLM. Investigator was blinded to genotype before measurements, but *egl-20* mutants were clearly distinguishable from the other genotypes. Statistical tests for differences between control (Bristol N2) and tagged EGL-20 or EGL-20 mutant strains were performed using a one-way ANOVA with Sidak's multiple comparisons test. Fluorescence intensity profile values for tagged EGL-20 and *Pegl-20*>2x mKate2::PH were obtained in FIJI (*Schindelin et al., 2012*) by drawing a line the width of the worm from head to tail and using the 'plot profile' function. Fluorescence intensity values for graphs were calculated by subtracting off-worm background in a nearby region from the raw pixel intensities. Where necessary for comparing shapes of fluorescence intensity profiles, intensity data were normalized in order to plot profiles of differing absolute intensities on the same graph. For kymographs of FRAP recovery, we drew a line the width of the worm along the anteroposterior axis including the photobleached and nearby unbleached regions and used the KymoResliceWide plugin in FIJI (*Schindelin et al., 2012*) to plot average intensity along the line as a function of time. Wnt/EGL-20::YPET levels outside of body wall muscle were calculated by measuring the mean pixel intensity of a region of interest anterior to the *egl-20*-expressing cells that did not include body wall muscles marked by a *Pmyo-3*-driven transgene. Control and Morphotrap worms were imaged with identical settings on the same slides, and off-worm background in a nearby region was subtracted from the pixel intensities before analyses. A Mann-Whitney test was used to assess statistical significance. The direction of Q neuroblast migration in Morphotrap and control animals was scored by visualizing migrating cells using a *Pwrt-2* >2x

mKate2::PH marker in L1 stage animals, and a Fisher's exact test was used to assess statistical significance of migration reversals. All statistical tests were performed using Prism seven software (GraphPad Software).

### Reagent availability

Strains generated for this work will be made available through the Caenorhabditis Genetics Center (CGC). Correspondence and requests for other materials should be addressed to A.M.P.

## Acknowledgements

We thank Kacy Gordon, Christopher Lowe, and members of the Goldstein lab for discussions and critical reading of the manuscript. We thank Jennifer Heppert for insightful discussions throughout the course of this work and Daniel Dickinson for technical advice. The P*myo-3*>mCherry::PH strain was a gift of Lara Linden and David Sherwood. Some strains were provided by the CGC, which is funded by the NIH Office of Research Infrastructure Programs (P40 OD010440. This work was supported by NIH grant R01 GM083071 (BG), NIH post-doctoral fellowship F32 GM115151 (AMP.), and American Cancer Society post-doctoral fellowship PF-16–030 DDC (AMP).

## Additional information

### Funding

| Funder | Grant reference number | Author |
| --- | --- | --- |
| National Institute of General Medical Sciences | R01 GM083071 | Bob Goldstein |
| National Institute of General Medical Sciences | F32 GM115151 | Ariel M Pani |
| American Cancer Society | PF-16-030 DDC | Ariel M Pani |

The funders had no role in study design, data collection and interpretation, or the decision to submit the work for publication.

### Author contributions

Ariel M Pani, Conceptualization, Formal analysis, Funding acquisition, Investigation, Visualization, Methodology, Writing—original draft, Writing—review and editing; Bob Goldstein, Supervision, Funding acquisition, Methodology, Writing—review and editing

### Author ORCIDs

Ariel M Pani http://orcid.org/0000-0002-9338-9750
Bob Goldstein http://orcid.org/0000-0001-6961-675X

### Decision letter and Author response

Decision letter https://doi.org/10.7554/eLife.38325.028
Author response https://doi.org/10.7554/eLife.38325.029

## Additional files

### Supplementary files

• Supplementary file 1. PCR primers used to amplify transgene promoters.
DOI: https://doi.org/10.7554/eLife.38325.025

• Transparent reporting form
DOI: https://doi.org/10.7554/eLife.38325.026

## Data availability

All data generated or analyzed during this study are included in the manuscript and supporting files. Source data files have been provided for Figures 1 and 6.

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
