## [Decision Letter]

Thank you for submitting your article "Direct visualization of a native Wnt in vivo reveals that a long-range Wnt gradient forms by extracellular dispersal" for consideration by *eLife*. Your article has been reviewed by Marianne Bronner as the Senior Editor, a Reviewing Editor, and four reviewers. The following individuals involved in review of your submission have agreed to reveal their identity: Henrik Korswagen (Reviewer #1); Madelon Maurice (Reviewer #3).

The reviewers have discussed the reviews with one another and the Reviewing Editor has drafted this decision to help you prepare a revised submission.

Summary:

This manuscript describes an impressive combination of approaches to test whether Wnts can be dispersed from a local extracellular source. The work also tests how Wnt proteins move, and if they can be trapped at the cell surface. These are long standing issues in the field of developmental biology and the work will be of interest to a wide audience. We would welcome a re-submission that addresses the revisions described below. The reviewers are all positive and make minor suggestions to improve the manuscript as you will see in the detailed reviews below.

As editor, I don't see the need to make any major revisions. Reviewer 4 comments "The paper would be significantly strengthened if they could show that retention of EGL-20 in producing cells gives the expected phenotype" but I believe that this would require extensive new experiments. Unless you would have data addressing this point already at hand, I do not see the this as a mandatory addition to the revision. Please address the revisions elaborated below.

Reviewer #1:

This manuscript addresses a long-standing debate in the Wnt field: can Wnt proteins disperse in the extracellular environment to form functional gradients or are they mostly short-range signaling molecules. Using a functional, endogenously tagged EGL-20/Wnt protein in *C. elegans*, the authors provide compelling evidence that it forms a long-range gradient through free extracellular dispersal. They show this through several approaches, including photobleaching experiments to examine dispersal dynamics and a trapping approach in which they prevent spreading by expressing an anti-GFP nanobody on body wall muscle cells. Moreover, using a tracing approach, they show that distant localization of EGL-20 is not the result of expression during earlier stages of development. This is a well thought out and important study, and given the current debate on this topic, it will be of wide interest.

Reviewer #2:

Pani and Goldstein use a tagged endogenous Wnt signal to show that Wnts can disperse extracellularly. The study combines an impressive combination of approaches to test whether Wnts can be dispersed from a local source, if they can move extracellularly, if they can move rapidly, and if they can be trapped at the cell surface. The answer to all these questions is yes. The paper will be of great interest to the Wnt and morphogen fields.

The data look convincing (but I might have missed intricacies of *C. elegans* biology).

1) My main reservation is the set-up of the paper. In particular, the tethered Wnt experiment (Vincent lab) shows that Wnt *can* act at a short range and fulfill many functions of wild-type Wnt, but it does not contradict the observation that Wnts *are* dispersed and form gradients. The experiments in this paper do not really address the point if tethered Wnt could do the job (although the Morphotrap experiment is a start to address this question).

2) The Discussion section seems a little opaque. For example, I do not understand the reasoning behind the proposal that source cells might produce excess ligand, and I don't see the evidence for active extracellular mixing.

Reviewer #3:

The main focus of this study is to investigate an important and much debated question in the Wnt field: if and how Wnts spread in extracellular space to mediate long-rang signalling. The authors employ an elegant in vivo method to tackle this longstanding issue. They perform (live) imaging of endogenous, fluorescently tagged Wnt in concert with the labelling of plasma membranes of sending and receiving cells. By careful characterization and manipulation of both source and responding cells as well as the movements of Wnt proteins in between, the authors make a number of important observations.

EGL-20/Wnt sending (cluster of posterior cells) and responding cells (Q neuroblasts) are positioned at a considerable distance in *C. elegans* and do not make any direct cell-cell contact, generating compelling evidence that Wnts spread through extracellular space. Furthermore, the morphological characterization of sending and receiving cells rules out the possibility that cytonemes or dynamic filopodia are responsible for transport of Wnt proteins between cells. In addition, by employing a genetic trick by which cells that expressed EGL-20/Wnt at any point during early development can be traced, the authors conclude that an earlier proposed 'autocrine/cellular memory mechanism' (Alexandre et al., 2014) does not apply for EGL-20 mediated signalling. When EGL-20/Wnt was expressed in farther tail cells not normally expressing this Wnt, Wnt proteins still reached their target cells (Q neuroblasts, seam cells, body wall muscle cells) over a range of 75 μm. Thus, native contact patterns do not seem necessary for the long-range spreading of Wnts.

Another important finding concerns the observation that bright punctae within the secreted Wnt gradient overlap with Wnt receptor expression on target cells. These results thus suggest that punctae mainly represent local Wnt-receptor accumulations and that the actual transported Wnt particles must be of much smaller size and remain invisible by confocal microscopy analysis. Their findings are consistent with models for morphogen gradient formation in which free dispersal of the ligand is hindered and shaped by binding to cell surface receptors.

Finally, the authors employ a nanobody-based Morphotrap approach to limit tagged Wnt movement by capturing extracellular EGL-20 to body wall muscle cells. This significantly alters the anteroposterior shape of the Wnt gradient and affected the migration of neuroblasts that now only could capture reduced levels of Wnt that trickled through the Morphotrap.

I think this is a well-focused, thorough and solid piece of work that provides important contributions (both technical and conceptual) to the broad community of morphogen signalling. I have no major or minor concerns to report.

Reviewer #4:

In this paper Pani and Goldstein address an important if somewhat controversial question. Do Wnt proteins spread with tissues? They build a good case to argue that EGL-20, a *C. elegans* Wnt does spread, most likely by diffusion in the extracellular space. Key to their result is a knock-in allele that expresses fluorescently tagged EGL-20. They show that this protein forms a gradient and is fully active. The authors provide good evidence that the transfer of tagged EGL-20 from producing to receiving cell does not require cell contacts nor is it likely to be mediated by cytonemes. They also rule out a lineage-dependent spreading mechanism. To identify and label *egl-20*-expressing cells, the authors generated a reporter with upstream sequences. They also generated a strain expressing flp from the endogenous locus to lineage trace *egl-20* expressing cells. The results confirm that *egl-20* is produced by a diverse set of posterior cells and 'received' by anterior body wall muscles. The intricacies of worm anatomy will be lost to the non-specialist but the authors seem to make a good case for non-contact dependent Wnt transfer. The authors also show convincing colocalisation with receptors in punctae suggesting that all punctal *egl-20* could represent endosomes forming upon engagement with receptors. Moreover, FRAP experiments suggest that *egl-20* spread rapidly, most likely by secretion.

This paper is well written, nicely documented and addresses an important question. As such it is a good contender for publication in *eLife*. I would suggest one key improvement. Although the authors have shown convincingly that Egl-20 spreads, they have not completely tested whether such spread is essential. The paper would be significantly strengthened if they could show that retention of EGL-20 in producing cells gives the expected phenotype. For this, they would need to show that Morphotrapped EGL-20 is signalling-competent and then express the Morphotrap specifically in the producing cells. Alternatively, they could replicate the *Drosophila* membrane-tethered Wingless experiment.

---

## [Author Response]

Reviewer #2:Pani and Goldstein use a tagged endogenous Wnt signal to show that Wnts can disperse extracellularly. The study combines an impressive combination of approaches to test whether Wnts can be dispersed from a local source, if they can move extracellularly, if they can move rapidly, and if they can be trapped at the cell surface. The answer to all these questions is yes. The paper will be of great interest to the Wnt and morphogen fields.The data look convincing (but I might have missed intricacies of C. elegans biology).1) My main reservation is the set-up of the paper. In particular, the tethered Wnt experiment (Vincent lab) shows that Wnt CAN act at a short range and fulfill many functions of wild-type Wnt, but it does not contradict the observation that Wnts ARE dispersed and form gradients. The experiments in this paper do not really address the point if tethered Wnt could do the job (although the Morphotrap experiment is a start to address this question).

We agree with the reviewer that the Vincent lab paper Alexandre, Baena-Lopez and Vincent, (2014) on membrane-tethered Wingless (Wg) was in essence a particularly elegant sufficiency experiment and did not contradict that Wnts typically disperse across cells to form gradients. Ultimately, the membrane-tethered Wingless paper proposed that requirements for long-range Wnt spreading in general should be revisited, which we have done here using *C. elegans*. We have now clarified the significance of the membrane-tethered Wingless experiments in the Introduction and our interpretation of how they relate to our findings in the Discussion section. Given the evolutionary conservation of the Wnt signaling pathway, it is likely that tethered Wnt/EGL-20 could substitute for the diffusible form in *C. elegans* if expressed in the right cells. However, there is a key difference between the two systems – Alexandre et al.’found that *Drosophila wingless* is broadly expressed at early stages in prospective wing blade cells that are later patterned by Wnt signaling, which they proposed could explain the ability of tethered Wntto mediate wing patterning as a consequence of earlier autocrine signaling. In contrast, our lineage tracing experiments using *flp::F2A* inserted at the *egl-20* locus showed that *egl-20* is never expressed in the majority of receiving cells or their neighbors, which makes a similar mechanism unlikely.

2) The Discussion section seems a little opaque. For example, I do not understand the reasoning behind the proposal that source cells might produce excess ligand, and I don't see the evidence for active extracellular mixing.

We agree there is not direct evidence for these hypotheses, and we have revised the Discussion section to make it clearer that this is speculative and not yet tested. We brought up these points as proposals for how free Wnt dispersal might be able to establish and maintain a long-range Wnt gradient in vivodespite theoretical constraints on the speed and distance of free morphogen diffusion. Our proposal that extracellular mixing could be involved in morphogen spreading is also speculative and intended to raise the possibility that cellular behaviors might physically affect dispersal of free signaling molecules in vivo, which has not been widely discussed in the literature.

Reviewer #4:[…] This paper is well written, nicely documented and addresses an important question. As such it is a good contender for publication in eLife. I would suggest one key improvement.Although the authors have shown convincingly that EGL-20 spreads, they have not completely tested whether such spread is essential. The paper would be significantly strengthened if they could show that retention of EGL-20 in producing cells gives the expected phenotype. For this, they would need to show that Morphotrapped EGL-20 is signalling-competent and then express the Morphotrap specifically in the producing cells. Alternatively, they could replicate the Drosophila membrane-tethered Wingless experiment.

We have attempted to retain tagged EGL-20 in producing cells using a ubiquitously-expressed Morphotrap but found that co-expressing Morphotrap and tagged EGL-20 led to intracellular accumulation of the protein that may represent impaired secretion. This result is now included in the revised manuscript (subsection “Extracellular spreading between cells mediates Wnt/EGL-20 signaling”) and in a new figure supplement (Figure 6—figure supplement 1). Though we did not retain EGL-20 specifically in producing cells, our finding that *Pmyo-3>*Morphotrap animals and *egl-20* mutants have similar defects in QL neuroblast migration demonstrates that free EGL-20 spreading is essential in at least one well-known developmental context. We have also added diagrams of the Morphotrap design and a schematic of effects on Wnt distribution and Q neuroblast migration in Figure 6B and 6I.